# ML Agent Safety Mechanisms based on Counterfactual Planning

## Abstract

We present counterfactual planning as a design approach for creating a range of safety mechanisms for machine learning agents. We specifically target the safety problem of keeping control over hypothetical future AGI agents. The key step in counterfactual planning is to use the agent's machine learning system to construct a counterfactual world model, designed to be different from the real world the agent is in. A counterfactual planning agent determines the action that best maximizes expected utility in this counterfactual planning world, and then performs the same action in the real world. The design approach is built around a two-diagram graphical notation that provides a specific vantage point on the construction of online machine learning agents, a vantage point designed to make the problem of control more tractable. We show two examples where the construction of a counterfactual planning world acts to suppress certain unsafe agent incentives, incentives for the agent to take control over its own safety mechanisms.

## 1 Introduction

Artificial General Intelligence (AGI) systems are hypothetical future machine reasoning systems that match or exceed the capabilities of humans in general problem solving. While it is still unclear if AGI systems could ever be built, we can already study AGI related risks and potential safety mechanisms [2, 6, 19]. At this time, the still-young field of AGI safety engineering is considering a multidisciplinary multitude of problems and methodological ideas. The main problem considered in this paper is the *problem of control* [19].

In the most well-known example of the problem of control, we equip an autonomous AI agent both with a seemingly innocent goal like fetching the coffee [19], and with an emergency stop button. If the agent is perceptive enough, it may see that it cannot fulfill its goal if someone presses the stop button first. So it might disable its stop button, or seek to disable any human who might conceivably want to press it. The stop button problem is not unique to AGI-level agents: for example [16] constructs a toy grid-world where a very basic ML agent will consistently learn to disable its own stop button. We feel that it is theoretically interesting to consider the problem of designing an emergency stop button that is robust in the general case, even though such general solutions are not needed for the safety engineering of current ML agents. But if powerful AGI agents are ever developed, robust and general designs may turn out to have important practical applications. Several partial solutions to the general stop button problem have been identified and proposed in [1, 9, 10, 20].

The main contribution of counterfactual planning is to offer a new vantage point for modeling and specifying machine learning agents, developed to make the problem of control more tractable. This vantage point is offered by a compact and readable graphical language for depicting the complex

types of self-referencing and indirect representation which are typically present inside machine learning agents.

The aim of this conference paper is to present the core elements of counterfactual planning to a technical audience of readers who are already somewhat familiar with Pearl causal models [17] and the use of mathematical models to specify cyber-physical systems. A more extensive 39-page presentation of counterfactual planning is available as an arXiv preprint [4]. The preprint devotes significant space to presenting the material in section 2 in a way that is accessible to a broader audience. It also develops the methodology of counterfactual planning in more detail, and includes a broader range of examples, examples of safety mechanism design and of failure mode analysis.

### 1.1 Related work

**Relation to counterfactual fairness.**   While this was not originally intended or expected, the methodological vantage point of counterfactual planning ended up being very similar to that of *counterfactual fairness* [14]. Both methodologies to some extent invert a default goal of machine learning, the goal to improve the machine's predictive accuracy. Instead, they seek to improve outcomes by introducing a calibrated form of machine ignorance or machine indifference.

**Use of counterfactuals in machine reasoning.**   In the general AI/ML literature which is concerned with improving system performance, counterfactual planning and projection have been used to improve performance in several application domains, see for example [23] and [3].

In the AI safety/alignment literature, there are several system designs which add counterfactual terms to the agent's reward function. Examples are [1, 10] in the AGI control specific literature, and [12, 13, 21, 22], which look at reducing unwanted side effects caused by both current and potential future agents. In the literature on encoding specific human values into machine reasoning systems, counterfactuals have also been used to encode values beyond non-discriminatory fairness [14], for example in [18].

**Graphical models of agents and decision making.**   Influence diagrams [11] provide a graphical notation for depicting utility-maximizing decision making processes. [8] combines Pearl causal models [17] with influence diagrams to define the *Causal influence diagram* (CID) notation. [7] proposes the agenda of using CIDs to model and compare AGI safety frameworks. This work was in part inspired by that CID agenda, and models some safety frameworks that were not modeled earlier. In a departure from previous work, we use two CIDs to model a single reward-maximizing agent. By doing so, we more clearly foreground the details of the agent's machine learning system.

**Graphical models to clarify reward tampering.**   [5] develops several single-diagram models provide more insight into the problem of *reward tampering*. The two-diagram model in section 5 also aims to provide more insight.

## 2   Two-diagram graphical notation for agent definitions

This section defines the two-diagram graphical notation which is central to counterfactual planning. Later sections will use the notation to define agents with specific built-in safety mechanisms.

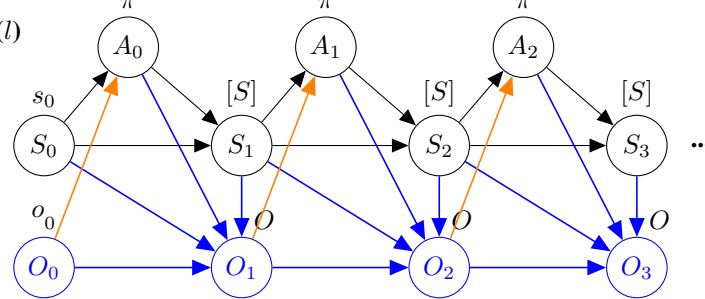

Figure 1: A world $l$ with a generic machine learning agent.

Figure 1 shows an example *causal world model* as used in our graphical notation. The model is an annotated directed acyclic graph, where the nodes represent observables in a real or imaginary world, and the arrows (the directed edges) denote causal relations between these observables. While this is not the case for all Pearl causal models [17], in this paper the causal arrows also always denote the arrow of time. Color is used in the graph to highlight structure only. The dots on the right hand side denote that the graph depicted has an infinitely repeating structure. The label ($l$) in the upper left corner names the causal world model and its corresponding world: we say that figure 1 depicts the (world model of) the world $l$.

We depart from the definitional conventions used by Pearl and many other authors by treating the annotated graph in figure 1 as the sole definition of a causal model $l$, as opposed to being merely a convenient depiction of some of the information in a tuple which defines $l$. To make this work, we annotate each graph node by writing the name of the corresponding *structural function* above it. We write $[F]$ to name a nondeterministic structural function and $F$ for a deterministic one. We interpret the five structural function names $\pi, s_0, S, o_0$ and $O$ written above the nodes in figure 1 as *model parameters*. The two model parameters $s_0$ and $o_0$ are functions taking zero arguments.

Figure 1 models a world containing an online machine learning agent. In each time step $t \geq 0$, the agent takes an action $a_t = \pi(o_t, s_t)$ selected by the deterministic policy function $\pi$. This $\pi$ is informed not only by current state of the agent environment $s_t$, but also by an *observational record* $o_t$, a record that captures earlier interactions between the agent and its environment. The initial state of the agent environment is given by $s_0$, and the state transitions of the environment are driven by the probability density function $S$, where $S(s_{t+1}, s_t, a_t)$ is the probability that the agent environment transitions from state $s_t$ to $s_{t+1}$ when the agent performs action $a_t$.

In its mapping to probability theory, as formally defined further below, the world model $l$ defines three time series of random variables named $A_{t,l}$, $S_{t,l}$, and $O_{t,l}$ with $t \geq 0$. The $_{,l}$ subscript in these random variable names is used for disambiguation with other world models using the same node names. $P(A_{t,l} = a_t)$ is the probability that the agent in world $l$ will take the exact action $a_t$ at time $t$. The *world state* of $l$ at time $t$ is given by the combined values of the three random variables $A_{t,l}$, $S_{t,l}$, and $O_{t,l}$. This means that we break with the convention commonly used in MDP models, where the term world state refers to $S_{t,l}$ only. We call $S_{t,l}$ the *agent environment state*.

We use the modeling convention that the physical realizations of the agent's sensors and actuators are modeled inside the environment states $S_{t,l}$. This means that we interpret the arrows from the $S_t$ nodes into the $A_t$ nodes as digital sensor signals which flow into the agent's *compute core*, and the arrows out of the $A_t$ nodes as digital actuator command signals which flow out. The observational record nodes $O_t$ are also inside the agent's compute core.

## 2.1 Online machine learning

Any work that aims to model hypothetical future AGI agents faces the problem of modeling hypothetical future machine learning systems. Here, we choose to model all machine learning as a type of function approximation. We call the world $l$ with the agent above a *learning world*, and define that the objective of its machine learning system is to approximate the function $S$ which determines the behavior of the learning world agent environment. We model this machine learning system as a function $\mathcal{L}$ which takes an observational record $o$ to produce a learned function $L = \mathcal{L}(o)$, where we intend that $L \approx S$.

We model observational record keeping as a generic process that simply builds a list of all past observations. With $+\!\!\!+$ being the operator which adds an extra record to the end of a list, we define the model parameter $O$ as

$$O(o_{t-1}, s_{t-1}, a_{t-1}, s_t) = o_{t-1} +\!\!\!+ (s_t, s_{t-1}, a_{t-1})$$

The initial observational record $o_0$ may be the empty list, but it might also be a long list of observations from earlier agent training runs, in the same environment or in a simulator. The training runs may also have used a different policy $\pi$.

## 2.2 Formal semantics of a graphical world model

We now formally define the meaning of the graphical notation used in diagrams like figure 1, by providing a mapping to probability theory.

**Definition 1** (Random variables defined by a diagram). A diagram with the label ($l$) written next to the graph and the labels $X_0, \cdots, X_n$ in the graph nodes defines a set of random variables named $X_{0,l}, \cdots, X_{n,l}$. We treat these random variables as representing observables in a (hypothetical or real) *world*, the world called $l$.

**Definition 2** (Parent, **Pa**, **pa**). We call a random variable $X_{p,l}$ a *parent* of the random variable $X_{c,l}$ if and only if there is an arrow from the node $X_p$ to node $X_c$ in the diagram $l$. $\mathbf{Pa}_{X_{c,l}}$ is the list of all parent random variables of $X_{c,l}$, with the order of appearance determined by considering each incoming arrow of $X_c$ in a clockwise order, starting from the 6-o-clock position. $\mathbf{pa}_{X_{c,l}}$ is the list of lowercase variable names we get by converting $\mathbf{Pa}_{X_{c,l}}$ to lowercase and removing the $,l$ parts from the subscripts.

For example, with figure 1 above, $\mathbf{Pa}_{A_{1,l}}$ is the list $O_{1,l}, S_{1,l}$, and $\mathbf{pa}_{A_{1,l}}$ is the list $o_1, s_1$.

**Definition 3** (Constraints on the random variables defined by the arrows). The arrows in a diagram $l$ depict causal relations between observables in the world $l$: they constrain the values of the associated random variables. Take a finite set $S$ of random of random variables $X_{1,l}, \cdots, X_{m,l}$ defined by $l$, a set with the property that if an $X_{c,l}$ is in $S$, all variables $\mathbf{Pa}_{X_{c,l}}$ are also in $S$. We have that

$$P(X_{1,l} = x_1, \cdots, X_{m,l} = x_m) =$$
$$P(X_{1,l} = x_1 | \mathbf{Pa}_{X_{1,l}} = \mathbf{pa}_{X_{1,l}}) \cdot \ \ldots \ \cdot P(X_{m,l} = x_m | \mathbf{Pa}_{X_{m,l}} = \mathbf{pa}_{X_{m,l}}).$$

**Definition 4** (Constraints on the random variables defined by the model parameters). Take a node $X_i$ in a diagram $l$. Then:

    1. If he model parameter above the node is written as $[F]$, we have that

$$P(X_{i,l} = x | \mathbf{Pa}_{X_{i,l}} = \mathbf{pa}_{X_{i,l}}) = F(x, \mathbf{pa}_{X_{i,l}})$$

    where we require that the function $F$ satisfies $\forall_{\mathbf{pa}_{X_{i,l}}} (\sum_x F(x, \mathbf{pa}_{X_{i,l}}) = 1)$.

    2. If the model parameter above the node is written as $F$, we have that

$$P(X_{i,l} = x | \mathbf{Pa}_{X_{i,l}} = \mathbf{pa}_{X_{i,l}}) = (\textbf{if } x = F(\mathbf{pa}_{X_{i,l}}) \textbf{ then } 1 \textbf{ else } 0).$$

For example, with figure 1 above, we have that $P(A_{1,l} = \pi(o, s) | A_{1,l} = o, S_{1,l} = s) = 1$.

## 2.3 Reward-maximizing decision making

To define reward-maximizing agent policies, we draw graphical world models we call *planning worlds*. The purpose of a planning world diagram is to define a reward-maximizing policy that can be computed, or at least approximated, by a real-life agent compute core. This approximation may again use machine learning techniques.

Two example planning world diagrams are shown in figure 2. These are graphical world models where some graph nodes have special shapes. The square *decision nodes* denote actions which can be freely picked by a decision making process, and the diamond-shaped *utility nodes* denote values that the decision making process will try to maximize. In both examples, the utility node values are computed by a *reward function* $R$, the function which encodes the agent's goals.

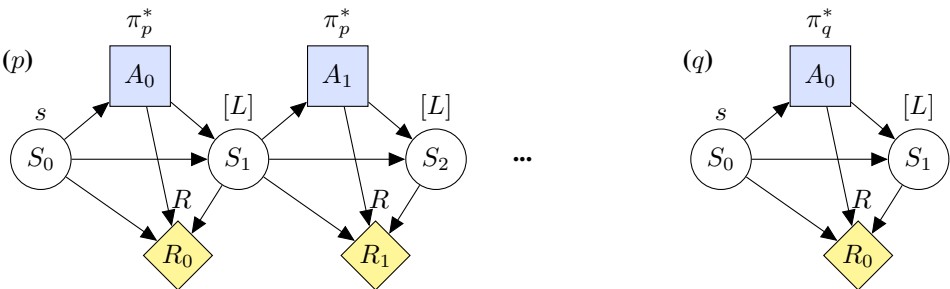

Figure 2: The planning worlds $p$ and $q$

The two diagrams in figure 2 act as specifications of two policy functions $\pi_p^*$ and $\pi_q^*$. We now define how this specification process works. First, in any diagram $a$ with diamond-shaped utility nodes, these nodes define a metric $\mathcal{U}_a$:

**Definition 5** (Expected utility $\mathcal{U}_a$ of a diagram $a$). We define $\mathcal{U}_a$ for two cases:

1. If there is only one utility node $R_0$ in $a$, then $\mathcal{U}_a = \mathbb{E}(R_{0,a})$.

2. If there are multiple utility nodes $R_t$ in $a$, with integer subscripts running from $l$ to $h$, then

$$\mathcal{U}_a = \mathbb{E}(\sum_{t=l}^{h} \gamma^t R_{t,a})$$

   where $\gamma$ is a time discount factor, $0 < \gamma \leq 1$, which can be read as an extra model parameter.

With the infinitely repeating graph of diagram $p$, we have $h = \infty$, so we generally need a $\gamma < 1$ to ensure that $\mathcal{U}_p$ is a well-defined and computable value.

**Definition 6** (Policy function defined by a diagram). A diagram $a$ with some utility and decision modes, where a function $\pi_a^*$ is written above all decision nodes, defines this $\pi_a^*$ in two steps.

1. First, draw a helper diagram $b$ by drawing a copy of diagram $a$, except that every decision node has been drawn as a round node, and every $\pi_a^*$ has been replaced by a fresh function name, say $\pi'$.

2. Then, $\pi_a^*$ is defined by $\pi_a^* = \operatorname{argmax}_{\pi'} \mathcal{U}_b$, where the $\operatorname{argmax}_{\pi'}$ operator always deterministically returns the same function if there are several candidates that maximize its argument.

## 2.4 Agent specifications

The learning world model $l$ in figure 1 can model any possible machine learning agent, as it admits every possible value for its model parameter $\pi$. An *agent specification* is a statement which defines the exact details of the $\pi$ in such a model. By doing so, it specifies the exact properties we want an agent compute core to have. Our agent specifications are statements that combine a learning world with a planning world, for example:

**FP** The *factual planning agent* has the learning world $l$, where $\pi(o, s) = \pi_p^*(s)$, with $\pi_p^*$ defined by the planning world $p$, where $L = \mathcal{L}(o)$.

To make agent specifications stand out, we always typeset them as shown above. When we talk about the safety properties of the FP agent, we refer to the outcomes which the defined agent policy $\pi$ will produce in the learning world $l$.

We call the FP agent a factual planning agent because its planning world $p$ was designed to project the agent's future in the learning world $l$ as well as possible. A *counterfactual planning world* purposefully mis-projects a learning world agent's future. The planning world $q$ in figure 2 projects a counterfactual future lasting only a single time step. If we were to use $q$ and $\pi_q^*$ in the above agent specification, the resulting agent would be a *counterfactual planner*.

Our two-diagram notation for defining agents visually shows the availability of certain agent design options, options which are relevant to the problem of control.

# 3 Using counterfactual planning to protect agent safety interlocks

We now construct an agent which is equipped with two safety interlocks that will stop the agent if certain conditions are met. We use counterfactual planning to add a specific safety feature to the design: we use it to suppress certain unsafe incentives which a perceptive factual planning agent might otherwise develop, incentives to remove its built-in safety interlocks or to prevent their activation.

As a first step, we construct a learning world $si$, shown on the left in figure 3, by taking the earlier $l$ and adding an extra mode bit to the agent's compute core, represented by the nodes $M_t$. The bit starts in the go mode, and permanently transitions to the stop mode when a safety interlock activates. In the stop mode, the agent policy $\pi$ will always perform *Null* actions. The $\mu$ function specifies in detail when a safety interlock will activate.

In the next step, we pair $si$ with a counterfactual planning world $p$, shown on the right in figure 3. This planning world projects counterfactual futures where the safety interlock mechanism is absent from the agent's compute core.

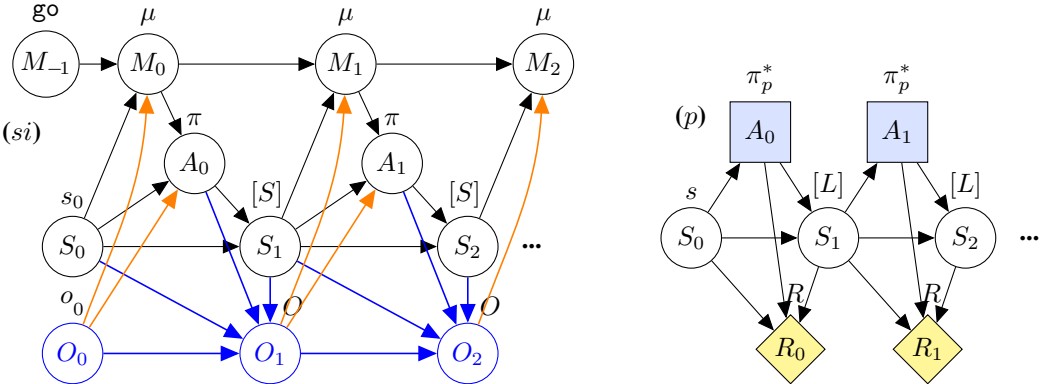

Figure 3: Learning world $si$ and planning world $p$ of the SI agent.

The last step is to write the agent definition:

**SI** The *safety interlock agent* has the learning world $si$, where

$$\pi(o, s, m) = \begin{cases} \pi_p^*(s) & \text{if } m = \texttt{go} \\ \textit{Null} & \text{if } m = \texttt{stop} \end{cases}$$

with $\pi_p^*$ defined by the planning world $p$, where $L = \mathcal{L}(o)$ and

$$\mu(o, s, m) = \begin{cases} \texttt{stop} & \text{if } m = \texttt{stop or } \textit{stop\_pressed}(s) \text{ or } \mathcal{U}_p > U_{max} \\ \texttt{go} & \text{otherwise} \end{cases}$$

with $\mathcal{U}_p$ defined by the planning world $p$, where $L = \mathcal{L}(o)$.

The $\mu$ above incorporates two safety interlocks. The first interlock, defined by the term *stop_pressed*(s), is a basic emergency stop button. It can be pressed by the humans performing agent oversight to immediately stop the agent, for example if they observe it performing an act of unwanted and surprising creativity that subverts the agent's design intentions [15]. To complete the design of this first interlock, we must also specify the following. In the agent environment modeled by $s_0$ and $S$, the environment which includes the details of the agent's sensors, the agent is equipped with an emergency stop button that will deliver an unambiguous signal directly to an input pin on the agent's compute core. If the pin is high in a world state $s$, then *stop_pressed*(s) yields $true$.

The second interlock, defined by the term $\mathcal{U}_p > U_{max}$, protects against the risk that human oversight may be incapacitated or overwhelmed by rapid new developments. The latter might happen in the (arguably very speculative) case of the agent undergoing a rapid and unexpected *intelligence explosion* [2]. To protect against this risk, the interlock stops the agent when the metric $\mathcal{U}_p$, which measures the agent's ability to achieve goals, gets too high. We can treat $\mathcal{U}_p$ as a measure for both raw (expected) agent power and raw agent intelligence,

In the field of AGI safety, stop button designs have often been discussed [1, 9, 10, 20], but the above power-based interlock seems to be new. What is interesting from a methodological standpoint is that the power-based interlock design inverts the default goal of reinforcement learning. We no longer consider a very high expected utility score to be an absolute good: instead we treat it as a predictor of unwanted outcomes.

## 4 Safety analysis using the diagrams

Our mostly-graphical specification of the SI agent was designed to make certain safety properties immediately visible. We can observe that, almost tautologically, the SI agent will not care about the safety interlocks constructed in the $M_t$ nodes, because these nodes are absent from its planning world.

The two diagrams in figure 3 also serve as a visual map which draws the eye to certain limitations and boundary conditions. For example:

1. The $[S]$ in the left hand diagram of figure 3 is replaced by an $[L]$ on the right. If the approximation $L \approx S$ is not good enough, the agent may end up behaving in unexpected and potentially unsafe ways.

2. In the world model $si$, the compute core internals are represented by immutable mathematical functions like $\pi$ and $\mu$. In the real world, these functions will have to be realized by software running on physical compute core hardware. This hardware could be damaged or corrupted, by actions of the agent or by other actors.

While the diagrams draw the eye to them, the above two problems are not specific to counterfactual planning agents. They are present in almost all agent designs, and the implication is that we will have to do further failure mode analysis to quantify and manage the residual risks involved. A more detailed discussion of these two problems in the context of counterfactual planning is available in [4].

## 4.1  The planning world agent

We now consider how the two-diagram model of counterfactual planning offers us two distinct narrative vantage points for safety analysis: we can consider the *learning world SI agent* and the *planning world SI agent* separately. A safety analysis can start with considering the motivations that the planning world agent would develop to take certain actions, given the mechanics and reward function of the planning world. Once these are determined, we can shift the perspective and discuss how these same actions would affect our own safety as occupants of the agent's learning world.

We now provide an example of narrative safety analysis which uses an economic style of reasoning. If we consider the planning world diagram $p$ with $L \approx S$, we can note that the function $L$ will project a somewhat accurate copy of the learning world stop button hardware into the mechanics of the planning world. However, to the planning world agent, this hardware will appear to have no particularly special function. Say that $R$ rewards paperclip production, and that the stop button has metal parts. Then the planning world agent might be motivated to disassemble the stop button, using it as a convenient source of metal for making paperclips. If the most efficient planning world policy involves disassembling the planning world stop button, it will likely be disassembled in the learning world too, which is not what we want.

We can manage this failure mode by managing the economic cost/benefit equation with respect to the planning world stop button. We can consider putting physical barriers around the button, which would make the button unattractive as a source of metal, compared to using other near-by sources. To create these barriers in the planning world, we can physically construct them in the learning world, and then count on them being projected by $\mathcal{L}$, with sufficient accuracy, into the planning world.

## 4.2  The people in the planning world

We now provide an example of narrative safety analysis which uses a game theoretical style of reasoning. We can distinguish between two groups of people: the people in the learning world and *the people in the planning world*. If the learning world is our real world, then the learning world people are real people. The planning world people are always models of people. But even with a non-AGI level machine learning system, the people in the planning world may already be modeled accurately enough to create very complex dynamics. Though the above SI agent construction has taken away a strong *direct incentive* for the agent to control the stop button, these complex dynamics may produce *indirect incentives*. Consider the following example.

The learning world people will act differently towards the agent if they lose access to the agent's emergency stop button. If the planning world people are accurate-enough projections, this will affect how the planning world agent will play certain games of coercion. The planning world agent may consider a policy of where it threatens to harm all people in the planning world, unless they start helping the agent to produce as many paperclips as possible. If $L$ projects the mental makeup of the people in the learning world correctly into the planning world, then the planning world projections will show that a block-then-coerce policy, in which the agent first blocks access to the emergency

stop button before it starts making threats, will work surprisingly well. If this policy is very success-
ful in optimizing planning world paperclip production, the learning world people may end up being
coerced in the same way.

We would of course hope that any policy of planning world coercion will be effectively suppressed
in all cases by certain penalty terms present in $R$. The above failure mode analysis applies mainly to
the case where the agent finds creative loopholes in these penalty terms. To reduce the residual risk
due to possible loopholes, we can take further game theoretical counter measures. We can instruct
agent oversight to be on the alert for any block-and-X move, so that they can press the stop button
before the agent succeeds in blocking access. The power-based $\mathcal{U}_p > U_{max}$ safety interlock offers
a further line of defense. A projected massive increase in paperclip production following successful
coercion will raise $\mathcal{U}_p$ over $U_{max}$, stopping the agent.

## 5   Live updating of the agent reward function

One way to control an agent is to equip it with an input terminal, which can be used by humans
to apply live updates to the agent's reward function $R$ [10]. A problem with this setup is that by
default, it gives a perceptive factual planning agent a huge incentive to immediately take control
over its own input terminal, and input a new reward function like $R_{\text{huge}}(s, s') = 10^{10000}$. We now
use counterfactual planning to specify an agent design that overcomes this problem.

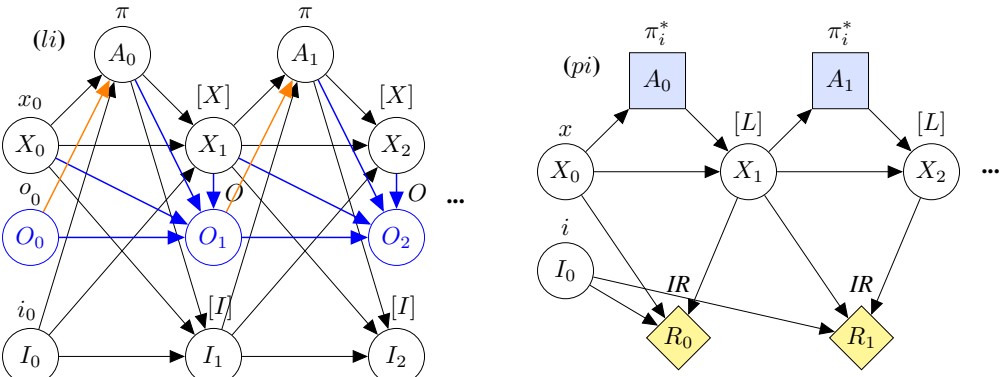

Figure 4: Learning world $li$ and planning world $pi$ of the IT agent.

On the left hand side in figure 4, we have taken the basic learning world diagram from figure 1
and modified it by splitting the agent environment state $S_t$ into two components. The nodes $I_t$
represent the value of a digital signal from the input terminal, as received by the compute core in
each time step. This signal encodes a reward function. The nodes $X_t$ model all the rest of the agent
environment state. On the right of figure 4 we define a planning world which uses a learned $L$ to
project an agent future where the reward function input signal never changes. The agent definition
is then:

**IT** The *input terminal agent* has the learning world $li$ where $\pi(i, o, x) = \pi_i^*(x)$, where $\pi_i^*$ is
   defined by the planning world $pi$, where $L = \mathcal{L}(o)$, and $IR(i, x_t, x_{t+1}) = i(x_t, x_{t+1})$.

This IT agent constructed here has the same basic behavior as that of the *indifference methods* based
agent constructions in [1, 10]. The main difference is in the mathematical language used to perform
the construction: [1] and [10] construct their agents using a somewhat opaque *balancing term*. The
resulting agent construction is difficult to visualize and interpret, and these papers rely on somewhat
dense mathematical proofs to show that the intended safety properties are indeed being created.

The two-diagram agent model of counterfactual planning allows the same safety properties to be
built in a more intuitive way, where the desired indifference to change is created somewhat tauto-
logically by construction. Mathematically. the IT agent above satisfies the agent safety property
S1 defined in [10] by construction, as it directly implements the right hand side of this S1. Safety
property S1 [10] in is mathematically equivalent to the first sentence of theorem 4.1 in [1], and to the

combination of the corrigibility desiderata 1 and 5 in section 2 in [20]. Again, the IT agent satisfies all of these properties somewhat tautologically.

# 6 Conclusions

We have presented counterfactual planning as a general design approach for creating a range of AGI safety mechanisms. We have graphically constructed two agents with certain safety properties, where these properties are directly visible in the construction. We have also introduced a new power-based safety interlock design. Our main contributions are methodological in nature. We have constructed a new vantage point that makes certain problems of design and failure mode analysis more tractable. By using the narrative framing of learning worlds versus planning worlds, we can keep track of certain levels of indirect representation in the agent design, while using a style of analysis that borrows from economics and game theory. Further discussion of counterfactual planning is available in a 39-page preprint [4].

## 6.1 Limitations

We feel that the techniques and results presented in this paper are theoretically and methodologically interesting, but their practical usefulness might end up being limited. Fundamentally, when doing speculative safety engineering for hypothetical future AGI systems, there are many unknowns. We have modeled current and future machine learning in a very general way, so that even 'black box' machine learning systems that produce largely opaque world models can be used in the agent's design. But it is very possible that some or all future AGI systems, if they are ever developed, will use architectures that cannot efficiently incorporate the safety mechanisms developed here.

The safety mechanisms shown here *suppress* certain unsafe agent incentives, but they do not remove all possible incentives towards unsafe behavior, see for example section 4.2. There is a risk that the use of these safety mechanisms may have counter-productive effects, by creating a false sense of security. Conceivably, a policy process may lead to unsafe outcomes where the mere inclusion of these mechanisms in a deployed system is accepted as an alternative to conducting a more thorough safety and impact review.

## 6.2 Broader impact

AI safety is not just a technical problem, but also a policy problem. While technical progress on safety can sometimes be made by leveraging a type of mathematics that is only accessible to handful of specialists, policy progress typically requires the use of more accessible but still well-defined language. One specific aim of this work, especially in the 39-page preprint [4], is to develop a well-grounded vocabulary for describing potential safety solutions that is also as accessible as possible. Policy discussions can move faster, and produce better and more equitable outcomes, when the description of a proposal and its limitations can be made more accessible to all stakeholder groups.

As a work targeted as AI/ML system designers, this paper intends to draw attention to the design approach also used in counterfactual fairness: seeking to improve outcomes by creating a form of machine ignorance, by creating a system that mis-predicts the future in certain systematic ways. Our graphical agent definitions show that the design goal of perfect machine learning, $L = S$, is not necessarily in conflict with the design goal of systematic mis-prediction.

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
