# OpenReview forum: "ML Agent Safety Mechanisms based on Counterfactual Planning"
_NeurIPS.cc/2021/Conference — NeurIPS 2021 Submitted_

### Official Review · Reviewer_QxvQ · 2021-07-16

**Rating:** 4
**Confidence:** 4

**Summary:**

This paper presents an approach to AI safety based on graphical models. It argues that it is providing new insights to safety problems which leverage these models. The paper exemplifies this by using the case of a stop button, and demonstrates how alterations to the graphical model (stop button, input terminal) in the learning (i.e real) world but not the planning (i.e. agent training/reasoning) allows safety to be introduced and not circumvented during the learning process modelled in the planning world.

**Limitations And Societal Impact:**

The limitations section is honest and enjoyable to read

**Main Review:**

The positive of the work is that using a graphical language for two models (learning, planning) for presenting, motivating, and discussing these cases removes the reliance on formal mathematical semantics or informal linguistic semantics when considering hypothetical future AI agents. This can reduce the ambiguity and sleight of hand of these kinds of discussions particularly for researchers that cannot access the formal mathematics of some of the cited papers. This can broaden the field and simplify the discussion, which is a good thing.

Considering the negative, I will break the paper into 2 parts: the modelling of an AI system as a graphical model, and the use of this model to discuss AI safety.

On the modelling side, the approach is near identical to the work by Toussaint et al under the banner planning as probabilistic inference.
- https://argmin.lis.tu-berlin.de/papers/12-botvinick-TICS.pdf
- https://www.ics.uci.edu/~dechter/courses/ics-295/fall-2013/papers/toussaint-probinf-pomdp.pdf
etc.

This takes the same approach of modelling states, actions, observations etc. as random variables. It does not explicitly deal with causality, but it does use directed arrows for the progression of time. I suspect they would argue the arrows are causal too. I would not want to see this work accepted for publication without a significant effort linking the proposed modelling formalism to Toussaint's work.

On the use of this model to analyse AI safety, whilst the clarity it brings is good, I fear it simply moves the casual response to AI safety ("don't let it access it's own stop button/reward function") from prose to a diagram ("we simply delete the arrows/factors from the planning world"). Since I am not an expert in this area I can't tell how much of an issue this is: the positive take (as above) is that it's clear from the diagrams, the negative take is that there is still lots of room for ungrounded hypothesising without further specification of a model. But since this is philosophy, that's part of the process.

Minor typos/issues:
- "If he model parameter above"
- Why use S as the transition function, not T as is standard elsewhere?
- I don't get the "digital actuator" part of the mode design. Does this mean A_0 only affects the physical state in S_2? i.e. there is a digital effect in S_1 then a physical effect in S_2?

**Time Spent Reviewing:**

1.5

---

> ### Author Response · Authors · 2021-08-09
> **On Toussaint et al taking the same approach to graphical modeling**
>
> The reviewer mentions that Toussaint et al take the same approach to graphical modeling as used by the CIDs (Causal Influence Diagrams) as defined and shown in the paper.  On a general level, we agree with this observation.  We were not aware specifically of the existence of the work by Toussaint et al, so we thank the reviewer for pointing it out.
>
> However, the reviewer also proceeds to say that "I would not want to see this work accepted for publication without a significant effort linking the proposed modeling formalism to Toussaint's work."  We recognize that this is a strongly framed request, and we would like to explore what we might do to make the reviewer feel we satisfied it.
>
> We admit that we are somewhat struggling to understand where the strong concerns of the reviewer lie.  We can imagine that their comments might flow from the reviewer being either a) strongly concerned that we are over-stating the newness of our approach, or b) strongly concerned that by failing to take the specific contributions to the field made by Toussaint et al into account, we have missed major opportunities to improve or validate our own research contributions.  Or maybe both, or none of the above.
>
> We will now offer some comments on these issues, while also including some questions that we hope the reviewer can answer during the submissions discussion period of August 10 – September 2, 2021.
>
> First off, we want to clarify what type of claim of novelty we intend to make for the submitted paper, when it comes to the paper introducing a new graphical modeling formalism.  What we intend to claim is that, compared to previous work (excluding the 39-page preprint), the methodology of using two diagrams to define and model a single agent, two diagrams representing a learning world and a planning world, offers a new vantage point.  We note that the linked-to work of Toussaint et al does not use its diagrams in this way, to clarify the distinction between learning and planning worlds.
>
> Also, we definitely did not want to claim or imply that CIDs, as a graphical modeling approach, are a modeling approach that is particularly novel.  For example, CIDs are very similar to the dynamic decision network (DDN) picture that is shown in the standard textbook 'Artificial intelligence: A modern approach' by Russell and Norvig. (A web search shows that this DDN picture was already present in the 2003 edition of this book.  In the edition we have in front of us here, a picture of a DDN appears as figure 17.3 in section 17.4.3.)  The historical notes at the end of chapter 17 of R&N mention that DDNs already appeared in the book Planning and Control by Dean and Wellman (1991), though we have not been able to find out if this book also used a similar graphical representation.  The works by Toussaint et al linked to by the review are from 2006 and 2012, and these two linked papers make no claim that the graphical notation being used in them is particularly new.
>
> CIDs, as a name for a graphical notation, were introduced by Everitt et al around 2019: see https://arxiv.org/abs/1902.09980 .  One might claim that the name CID is merely a new name for a pre-existing graphical notation.  But we feel that the introduction of the new name CID has value because it refers to a specifically well-defined version of that notation with specifically well-defined probabilistic semantics, while also referring to an agenda to use this notation as a standard tool for AI/AGI safety analysis.
>
> If the reviewer feels that our submission includes text that implies that CIDs are completely novel as a graphical modeling notation, then please point out the specific lines that create that impression, because we will definitely want to rewrite them to remove that implication.
>
> It was not our goal, in the related work section of the submission, to include a complete history of the use of CID-like/DDN-like graphical models of agency or decision making.  Given our description of the state of the field above, we hope that the reviewers will appreciate that such a complete history would likely have to be very long.
>
> That being said, there is room in the paper (given the extra page allowed for taking reviewer comments into account) to add more references to other work that used graphical models: if a reviewer feels strongly that adding a particular extra reference would be highly useful for some or all of the intended readers, we are definitely willing to follow up on that insight.
>
> We will now take the analytical stance of treating the two papers of Toussaint et al mentioned by the reviewer as suggested candidates for inclusion in the related work section of the submitted paper.  We observe the following.
>
> Looking at 'Planning as inference'.  https://argmin.lis.tu-berlin.de/papers/12-botvinick-TICS.pdf , this is a paper that constructs a graphical model to serve as a theory of how human planning might work as a neurophysical process.  The paper advances the theory that humans build probabilistic cognitive models inside their minds: 'Under the planning-as-inference (PAI) view, the decision-making agent makes use of an internal cognitive model, which represents the future as a joint probability distribution over actions, outcome states, and rewards'.  This internal cognitive model is very similar (both graphically and mathematically) to our concept of a planning world that can be built inside an agent compute core.  So we might observe that the model of agent cognition we use in counterfactual planning has a lot of similarities to the model of human cognition in 'Planning as inference'.
>
> This observation may be interesting to some or many readers in the NeurIPS audience: a good case could be made that it would be useful to discuss this similarity in the related work section of the submission, where a reference to 'Planning as inference' could then be usefully included.  Our question to the reviewer is if they would support that case, based on their knowledge of the NeurIPS audience.
>
> We note however that the theory (or analytical stance) that the human might is capable of building probabilistic models is not a novel contribution to the field that is made by 'Planning as inference': this theory and stance has historically been common in philosophy, linguistics, and presumably also neuroscience.  So we feel that the case for citing 'Planning as inference' specifically, as compared to some other source, is not very compelling.  This in turn makes us wonder if we may be missing some crucial insight that the reviewer intended to communicate to us.
>
> We note that the novel contribution claimed in the 'Planning as inference' is to further investigate the 'still unanswered question: how exactly does planning happen [in the human brain]?'.  The parallel question for AI is: how exactly might an agent the agent 'solve' the planning world model, to obtain or approximate the optimal policy $pi^*_p$ defined by the planning world, or at least an approximation of the value $pi^*_p(s)$ that is mentioned in the SI agent definition?  A comparative discussion the many different possible algorithms/approaches that might be used by machine minds or by human mind to 'solve' a planning world model is outside the intended scope of our paper.
>
> Looking at 'Probabilistic inference for solving (PO)MDPs', https://www.ics.uci.edu/~dechter/courses/ics-295/fall-2013/papers/toussaint-probinf-pomdp.pdf , we note that this is a paper presents a method for 'solving' MDPs or POMDPs, by mapping these to a structured dynamic Bayesian network (DBN) and applying a specific set of tools to these mappings.  In the terminology of our submission, this paper by Toussaint et al offers a specific technique for or obtaining or approximating the $pi^*_p(s)$ as defined by a planning world.  While the technique shown in the paper by Toussaint et al is interesting, again a comparative analysis of techniques for 'solving' a planning world model is outside the intended scope of our paper. So we do not feel that there is a compelling case for our paper to discuss or cite 'Probabilistic inference for solving (PO)MDPs'.
>
> Again, it is possible that we overlooked some detail in 'Probabilistic inference for solving (PO)MDPs' that offers a compelling case for us engaging with it: if so we ask the reviewer to point it out.

---

> > ### Comment · Reviewer_QxvQ · 2021-08-23
> > **On Toussaint et al taking the same approach to graphical modeling**
> >
> > Thanks for your detailed response. I chose those two papers as exemplars from a wider body of work, and chose two that I hoped would give you a good way in to planning as inference.
> >
> > > We admit that we are somewhat struggling to understand where the strong concerns of the reviewer lie. We can imagine that their comments might flow from the reviewer being either a) strongly concerned that we are over-stating the newness of our approach, or b) strongly concerned that by failing to take the specific contributions to the field made by Toussaint et al into account, we have missed major opportunities to improve or validate our own research contributions. Or maybe both, or none of the above.
> >
> > A little from column A, a little from column B. Since I see your CIDs as building on model which is more-or-less the same model used in PAI, I believe it is important that you include this link for context, just as you reference other similar graphical models. This is as much for future readers as for you. If someone wants to start researching how to automatically infer actions from CIDs (e.g. for safety verfication), then they need this context. I also expect that results from the PAI literature can help you understand what decision-making tasks can and cannot be modelled (easily) in this way.

---

> > > ### Author Response · Authors · 2021-08-26
> > > **Thank you**
> > >
> > > Thank you! Your clarification resolves our questions about the nature
> > > of your concerns.

---

### Official Review · Reviewer_NWTJ · 2021-07-16

**Rating:** 4
**Confidence:** 3

**Summary:**

This paper present counterfactual planning as an approach for creating safety mechanisms for agents.  It is framed in the context of AGI.  The topic is relevant to the NeurIPS community and the paper is interesting and well written.  I do believe there are contributions in this work towards building safer AI systems and the way that it proposes to do so.  Other than those concerns articulated in my main review below, I think the work is thorough and well presented.  The technical correctness and detail seems sufficient.

**Ethical Concerns:**

I don't have any ethical concerns with the work.  There were no human subjects experiments, or use of data that could be considered unethical.

**Limitations And Societal Impact:**

This is a theoretical paper which the authors admit may or may not have implications.  The motivation is a good one and they do discuss the broader impact and societal implications of their work.  I don't have any concerns there.

**Main Review:**

Below I will list my concerns about the paper and the questions for the authors. Unfortunately the first might be something that is difficult to address in a rebuttal, but it would be helpful to have a little clarification or justification if the authors believe I have misunderstood things.

1. The authors reference the 39-page preprint explicitly in several places.  Unfortunately, I don't believe they have correctly followed the anonymization policy for NeurIPS which states " you need to cite one of your own papers, you should do so with adequate anonymization to preserve double-blind reviewing.  For instance, write “In the previous work of Smith et al. [1]…” rather than “In our previous work [1]...”)."
I can only assume that the report is by the same author and it should have been referenced in the manner above.  Related to this I was struggling to distinguish from the references to this paper, how the authors were distinguishing their submission from the pre-print. I'd like to have a clearer articulation of that in the rebuttal.

2. Perhaps my main concern with the paper is the illustrated by the lack of evaluation of the methodology.  The authors identify that their contribution is "at its core a theoretical one" but that they are primarily presenting a ``new models and methodologies'' which I would agree with.  It would have been nice to see an example of they models/methodologies demonstrated even on a toy example. What was the reasoning for not doing so?  I don't think these need to be extensive or elaborate but I think it would add a lot to the contribution. I wonder if this might help also differentiate this contribution from the 39-page preprint (albeit that I wasn't able to read that preprint in full).

3. From my perspective the related work sections was a little weak. I would have liked to have seen a little more discussion of the related work in counterfactual reasoning and other related areas. The were desperately short on space in their submission and so it would be nice to see a little more in the way of background.

I will note that my score below was influenced by the lack of anonymization.  I wasn't sure how much weight to give that.  Overall, I think the paper is otherwise generally good. But I was trying to follow the NeurIPS guidelines as closely as possible.

**Time Spent Reviewing:**

3

---

> ### Author Response · Authors · 2021-08-09
> **Clarifying the status of the referenced 39-page preprint**
>
> As requested, here is a clearer articulation of the status of the 39-page preprint referenced in the submission, and a discussion of how the authors applied the NeurIPS anonymization policies to the submission.
>
> As mentioned in lines 38-44 of the submitted paper, our core aim in writing this 9-page paper was to present, to a specific audience, highlights from the same research effort on counterfactual planning that produced the 39-page preprint.  For context: this preprint was already available online at the time the paper was submitted.
>
> It was not our goal for the 9-page paper to present additional new results beyond those already included in the 39-page preprint. (That being said, one could argue that the more condensed presentation, especially in section 2, has a degree of novelty in itself.)  Now, NeurIPS invites 'submissions presenting new and original research' only.  However, its criterion for newness of the research submitted does allow for the prior publication of this research in preprints.  To quote: 'The existence of non-anonymous preprints (on arXiv or other online repositories, personal websites, social media) will not result in rejection.'.  We therefore felt that, even given the existence of the 39-page preprint, we could prepare and submit a 9-page paper under the conference rules regarding newness, and further felt that doing so would be of value to the community.
>
> What we intended to happen in the review process of the 9-page submission was that all reviewers would judge the research results in this paper (on newness and other criteria) while disregarding the existence of the 39-page preprint, specifically while never even searching for, let alone reading, the 39-page preprint in question.
>
> Regrettably, in the case of reviewer NWTJ, this is not what happened.  They mention that 'I wasn't able to read that preprint in full'.  This implies that apparently, they did a web search for the title of the 39-page preprint, the preprint title on line 370 in reference [4] of the submission, and it implies that they successfully found the 39-page preprint online.  As this preprint was published on the web non-anonymously, this had the side effect of breaking the intended double-blind nature of the review process.
>
> One may consider, in retrospect, that these events probably would not have happened if the 9-page submitted paper had not mentioned or referenced the 39-page preprint at all.  However, when writing the submission we felt that only logical course of action possible was to include this reference, that adding this reference would be valuable to the intended readers of the 9-page paper.  We also felt that we had to include a honest discussion (on lines 38-44) about the exact relation between the 9-page paper and the 39-page preprint.  Omitting this reference and this discussion would have been a disservice to future readers.  On a more general level of paper quality it would also have created a large and avoidable hole in the discussion of related work.
>
> Given the above background, we now want to address the assessment by the reviewer that 'I don't believe they [the author(s)] have correctly followed the anonymization policy for NeurIPS.'.  Now, the policy says that 'If you need to cite one of your own papers, you should do so with adequate anonymization to preserve double-blind reviewing.'  The author instructions provide two examples of how one might perform this adequate anonymization.
>
> For reference, we now cite these two examples of adequate anonymization in full: <<For instance, write “In the previous work of Smith et al. [1]…” rather than “In our previous work [1]...”). If you need to cite one of your own papers that is in submission to NeurIPS and not available as a non-anonymous preprint, then include a copy of the cited submission in the supplementary material and write “Anonymous et al. [1] concurrently show...”).'>>
>
> We will remain silent on the question of whether we applied the first example approach, citing own previous work without saying 'our work'.  If we were to say here that we applied this example approach too, we would be de-anonymizing some or all of ourselves, which should not happen in a double-blind review.
>
> What we can say is that we felt that we could not apply this example approach to the adequate de-anonymization of the 39-page preprint referenced.  The text in lines 38-44 of the submission makes it clear, or at least strongly implies, that both the 9-page paper submitted and the 39-page preprint were produced by the same single research effort/team that created the methodology of counterfactual planning.  If we had included author names in the reference [4] of to the preprint, we therefore would not have adequately anonymized the identities or some or all of authors of the submission.  We therefore used the second approach suggested by NeurIPS, the use of the author names "Anonymous et al." in the reference [4] to the preprint.
>
> So overall, in light of the conference intent and instructions, we cannot agree with the feeling of the reviewer that we as authors have not correctly followed the anonymization policy of NeurIPS.  We feel that we have followed it in the best possible way that was available to us, in the context of writing a text that would be appropriately informative about related work.
>
> However, it is clear that de-anonymization did likely happen: reviewer NWTJ mentions that they located and read part of the 39-page preprint, and this 39-page preprint on the web has full author names on its title page.  This is all regrettable, but we feel that this is just one of these unintended de-anonymization events that can happen, even when all parties involved try their best to stop them from happening.  We of course regret that the event happened and offer our apologies for having played a part in it.  In retrospect maybe this outcome might have been avoided if we had included an additional temporary footnote in the text, addressed directly to the reviewers, to add an extra warning about the special status of reference [4].
>
> The reviewer mentions that this anonymization issue affected their scoring, so we invite the reviewer to consider if they want to make a change to their scoring.  We are also interested in the reviewer's answer to two specific questions:
>
> 1) in light of the above, does the reviewer still feel that the authors did not correctly follow the anonymization policy for NeurIPS.
>
> 2) the reviewer also mentioned that they were struggling with the question of "how the authors were distinguishing their submission from the pre-print".  Has the information provided above resolved this question, and if so, does this affect the reviewer's opinion on the strength, in terms of novelty of the results as expected by NeurIPS, of the material in the submitted paper?

---

> > ### Comment · Reviewer_NWTJ · 2021-08-21
> > **Response to authors.**
> >
> > I appreciate the reviewer's thoughtful response to the questions about anonymization.  In understand the challenges with anonymization when material is posted on Arxiv.  I'd encourage the authors to use the for "XXXX et al. [YY] report ..." even when referring to their own work.  Which I think generally helps make the reference feel more distanced from the current submission. In this case I am not really sure it was necessary to refer to the Arxiv post at all, in the paper should stand alone without the extended version that is available on ArXiv.  I felt as though that was a bit confusing as a reviewer.
> >
> > I didn't feel that the authors responded to my two other comments in my original review.  If they could provide responses to those that would be appreciated.

---

> > > ### Author Response · Authors · 2021-08-23
> > > **Response comments 2 and 3 by reviewer NWTJ**
> > >
> > > Below, we provide specific responses to comments 2 and 3 by Reviewer
> > > NWTJ, as the reviewer stated they would appreciate getting these
> > > responses.
> > >
> > > In comment 2, the reviewer notes that 'It would have been nice to see
> > > an example of [these] models/methodologies demonstrated even on a toy
> > > example.'  Now, sections 3, 4, and 5 do demonstrate the use of the
> > > methodology by constructing and analyzing two example (non-toy)
> > > agents.  So we read the above comment on 'toy example' as expressing a
> > > desire for the inclusion of a 'toy *world* example', e.g. an example
> > > demonstration of how the SI agent design would act in a toy world.
> > > For our discussion about including such examples, and the reasons why
> > > we did not do so in the paper, see our comment titled 'On improving
> > > the paper by adding more examples', as attached to the comments by
> > > reviewer F8kk.  We are interested in getting the reviewer's thoughts
> > > on this response.
> > >
> > > Comment 2 by reviewer NWTJ starts by mentioning a more generic
> > > concern: 'Perhaps my main concern with the paper is the illustrated by
> > > the lack of evaluation of the methodology.'.  We now consider this
> > > more generic concern, because we do not agree to the observation that
> > > the paper lacks evaluation of the methodology.  When we read
> > > 'evaluation' to mean 'a discussion that identifies the benefits and
> > > limitations of the methodology', the paper definitely includes this
> > > type of discussion.  If we read 'evaluation' more narrowly as 'a claim
> > > that there are benefits over earlier methods, preferably backed by
> > > metrics that can be checked independently', we note that this kind of
> > > evaluation is also present in the paper.
> > >
> > > Lines 313-324 examine how using counterfactual planning compares
> > > favorably to using the design approaches in [1] and [10].  Notably,
> > > this comparison is a fair and useful one, as [1] and [10] construct
> > > and analyze agents with the same safety properties as the IT agent in
> > > section 5.  (The cited [20] also aimed to design an agent with these
> > > same safety properties, but [20] is notable for reporting a negative
> > > design result. The authors of [20] failed to produce a design that met
> > > their desiderata and called for more research.)
> > >
> > > Lines 313-324 discuss how the earlier efforts [1][10] created agents
> > > using a construction which is 'difficult to visualize and interpret',
> > > compared to that of the IT agent.  So this is a first comparison
> > > between these different design methods: to state it in
> > > quantitative terms: we claim that the average reader will find the
> > > agent constructions in [1][10] more difficult to visualize and
> > > interpret than that of the IT agent.  This claim can of course be
> > > checked by the interested reader, if they are so inclined, by using
> > > themselves or others as a test subject attempting such visualization
> > > and interpretation.
> > >
> > > Lines 313-324 also identify a second metric: the amount of
> > > mathematical work needed to prove that the safety properties created
> > > exist.  The lines observe that the agent constructions in [1][10]
> > > require 'somewhat dense mathematical proofs to show that the intended
> > > safety properties are indeed being created'.  This need for dense
> > > proofs is contrasted with counterfactual planning as a methodology,
> > > where 'the desired indifference to change is created [proved] somewhat
> > > tautologically by construction.'  Again, the reader can check this
> > > difference, if so inclined, by examining the proofs in [1][10] and
> > > their role in the text.
> > >
> > > The reviewer's comment 3 states that the reviewer 'would have liked to
> > > have seen a little more discussion of the related work', and they
> > > clarify that they would appreciate a response on this comment.  We
> > > don't know if there is a very specific response the reviewer expects
> > > to get, we can only state the obvious response.  Obviously, we could
> > > have included more discussion in the related work section, but we did
> > > not do so because of space considerations: we prioritized other
> > > things.  However, we appreciate the feedback provided, this is
> > > valuable feedback in the context of writing future papers aimed at the
> > > NeurIPS community.

---

### Official Review · Reviewer_F8kk · 2021-07-16

**Rating:** 4
**Confidence:** 3

**Summary:**

This paper tackles the problem of designing "robust" safety mechanisms against hypothetical AGI agents that may override such safety mechanisms to complete its goal.
The paper proposes using counterfactual planning, where an agent plans in a world designed to be different from the real world. The key idea is to plan in a counterfactual world where the safety mechanism does not exist, and use such planned action in the real world. This can then reduce the likelihood of the agent learning a direct correlation between the safety mechanism and goal completion. The paper also argues such planning offers a new vantage point for modeling AGI agents due to its compact and readable graphical language. Two examples are presented to showcase the proposed design approach where construction of counterfactual worlds suppresses some unsafe agent incentives.


**Limitations And Societal Impact:**

The authors have adequately addressed the limitations and potential negative societal impact of their work.
Some comments on the inclusion of a power-based safety mechanisms can be helpful.

**Main Review:**

The paper proposes an interesting idea of planning in a counterfactual world to minimize the direct incentives for agents to learn to override safety mechanisms. This idea seems novel and technically sound.  The graphical modeling of AGI agents in its "learning world" and "planning world" helps with interpretability. The limitations listed in the paper are well-thought-out, especially on its potential unintended consequences such as, "false security".

To improve the paper, it'll be especially helpful to extend on Section 4 and 5 with well-thought-out and tangible examples. Currently, it is hard to evaluate the exact contribution of counterfactual planning to tackle practical examples. For more impact, it would be helpful to focus on one use case per section to thoroughly describe to readers how counterfactual planning can exactly counter the "stop button" problem in well-illustrated examples. For example, by describing what L and S should be for the paper clip example in line 259-264. These would help clarify what is the exact contribution of the proposed method to state-of-the-art methods. For example, how does counterfactual planning help users understand what motivates the planning world agent? It will also be very helpful if the paper includes a toy experiment with the paperclip example, to demonstrate exactly how counterfactual planning can discourage unsafe agent incentives.

Secondly, it will be helpful to offer more technical analysis on the impact of the assumptions and how violations of such assumptions can manifest. For example, on the assumption of L~S. It is unclear if this assumption is reasonable and how we can practically gauge this in the real world. As the paper mentions, "when this approximation is not good enough, the agent may end up behaving in an unexpected and potentially unsafe way". Examples can be given on what is such an unexpected manner and how big of an approximation gap can manifest such behaviors. As this paper mainly includes thought experiments, clarifying such assumptions and details can increase impact.

Thirdly, more details and analysis can be added to the power-based safety interlock mechanism. The paper claims this as a contribution, with such an interlock offering a further line of defense. However, more detail can be given to substantiate such claims with more examples, and also analysis on potential unintended consequences. For example, would imposing such a mechanism slow down learning?

Lastly, it will be helpful for some comments on how this idea can be implemented on real world systems. As part of the limitation, the authors do note that "their practical usefulness might end up being limited". Despite this, it'll be interesting to know what are potential test cases where this method can make the most impact.

With more thorough examples and analysis, I believe the findings of the paper can be significant to the community. The idea of planning in counterfactual worlds to design safety mechanisms seems sound and can be an interesting thought exercise when addressing AI safety and alignment.


**Time Spent Reviewing:**

7

---

> ### Author Response · Authors · 2021-08-09
> **On improving the paper by adding more examples**
>
> (This is a response to comments by several reviewers, and the text below has discussion questions addressed at all reviewers)
>
> Both reviewer F8kk and reviewer NWTJ suggest that the paper could be improved by adding more concrete examples, e.g. toy examples with a paperclip maximizer.
>
> First, we will answer a specific question from reviewer NWTJ: 'It would have been nice to see an example of they models/methodologies demonstrated even on a toy example. What was the reasoning for not doing so?'  Our main reason was lack of space, or more specifically that we wanted to use the available space to discuss other things.  A secondary reason was a certain lack of novelty.  Reference [10] of the paper includes several illustrative toy world simulations for an agent that is mathematically equivalent to the IT agent in section 5 of the submitted paper.  (We notice now that we failed to mention the availability of these illustrative simulations explicitly in submitted paper, so this is a point of improvement)
>
> Our objective and scoping for the submitted paper was to discuss counterfactual planning as a design approach/methodology.  As a design approach, counterfactual planning contains several interlocking parts: the graphical notation using two diagrams is one, but we consider the narrative safety analysis discussed in sections 4.1 and 4.2 to be an equally important part of the methodology.  The consequence of our scoping, when applied to a 9-page paper, was that we ended up discussing each part of the methodology only briefly, and that we treated the SI agent as running example, not as the central object under discussion,
>
> Based on the review comments, we are now considering abandoning the scoping we used, and instead preparing a differently scoped paper for future submission to NeurIPS or a conference like it.  Let's call this potential future paper P2 for the sake of discussion.
>
> This P2 would take the presentation of the SI agent design in section 3 as its central goal, not the presentation of the methodology that produced it.  P2 would also omit the material from sections 4,1, 4,2, and 5, creating enough space to present a specific illustrative toy example, an example that combines a toy world with the use or an existing ML algorithm that is strong enough to learn the full mechanics of the toy world. P2 would also include other points of clarification and elaboration about the SI agent, as suggested by the reviewers.
>
> The merit of P2 is that it would explain a single thing with a much higher level of detail, a level of detail that we understand the reviewers feel would add strength.  A demerit would be that P2 omits the presentation of the methodology that produced the SI design, and the material in sections 4.1 and 4.2, and 5.
>
> We have three questions to the reviewers:
>
> Based on this description, would they rank the potential paper P2 more highly than the current submission?
>
> Would they feel it would be valuable if we were to write this paper P2 and submit it to a high-impact general AI/ML conference like NeurIPS?
>
> Our third question relates to the potential inclusion of learning curves in P2, their inclusion as an explanatory device.  We observe that in the most archetypal modern AI/ML paper, the central claim that the design under consideration improves things over a baseline will always be backed up by a comparison between two learning curves.  We could potentially include such a comparison in P2.
>
> We feel however that, for the problem at hand, there are several comparative metrics which are more revealing (while taking less space) than learning curves.  So we are considering whether P2 should take the space to include both learning curves and these more specialized metrics.  The case for including learning curves too is that they make the P2 paper stronger by being a communicative device that many readers will expect and appreciate if it is included,
>
> Do any of the reviewers have a strong opinion about this case for including learning curves in P2?

---

> > ### Comment · Reviewer_F8kk · 2021-09-01
> > **Response to authors**
> >
> >
> > Thank you for the author’s response. Replying to the authors' question on whether the scope of proposed P2 or current P1, I believe it is the author’s call on the key differentiator of the paper and associated insights.
> >
> > From my understanding, this paper is to promote a new design methodology that (1) can lessen the stop button problem and (2) offer analysis not possible with current methods. I believe it’ll be helpful to showcase these impacts through deep discussions of one running example (Chapter 3,4 on stop button problem), especially while pointing out the limitations of current methods. Ideally, by focusing on one running example, there can be more analysis, such as on “the impact of assumptions and how violations can manifest” that is brought up in the original review.

---

> > > ### Author Response · Authors · 2021-09-01
> > > **Thank you**
> > >
> > > We thank the reviewer for commenting on the idea of a P2
> > > paper. Overall, our general feeling from the feedback is that this
> > > paper P2 would not necessarily be better received.
> > >
> > > We now include some technical comments, as the reviewer re-iterates
> > > their interest in a specific topic:
> > >
> > > > For example, on the assumption of L~S. It is unclear if this
> > > > assumption is reasonable and how we can practically gauge this in the
> > > > real world.
> > >
> > > Generally speaking, the reasonableness of the assumption L~S is
> > > closely related to the problem of out-of-distribution robustness.  It
> > > is a well known risk that, if an agent ever enters a particularly
> > > tough out-of-training-distribution environment, a lack of out of
> > > out-of-distribution robustness might act to defeat all the good
> > > intentions encoded into the agent's reward function.  We can extend
> > > this reasoning: a lack of out-of-distribution robustness in L may also
> > > defeat all the good intentions encoded into the agent's counterfactual
> > > planning based safety mechanisms.
> > >
> > > It is of course very possible to gauge a machine learning system for
> > > out-of-distribution robustness by benchmarking it.  That being said,
> > > in terms of high-reliability safety engineering, we face the
> > > problem that a lack of test failures on a finite
> > > out-of-training-distribution test set can never fully prove the total
> > > absence of failures in every possible deployment situation. This problem cannot
> > > be cleanly solved in a mathematical sense, so the real challenge is to
> > > manage it as well as possible, and this is a somewhat inexact science.
> > > Improving out of distribution robustness, and verifying out of
> > > distribution robustness for high-risk systems, are of course active
> > > areas of ML research.
> > >
> > > Counterfactual planning, as a design methodology, offers no new
> > > insights that could contribute to out-of-distribution robustness
> > > research. It does offer further motivation to conduct such research.  As mentioned in lines 244-245: 'While the diagrams draw the eye to them,
> > > the above two problems are not specific to counterfactual planning
> > > agents.'

---

### Official Review · Reviewer_opsT · 2021-07-17

**Rating:** 3
**Confidence:** 4

**Summary:**

This paper uses causal influence diagrams (CIDs) to consider abstract agents that plan according to a model of the world that lacks certain features, such as emergency stop buttons.  As the model used for planning doesn't contain the stop button, the agent will have no incentive to ever remove this.

**Ethical Concerns:**

No concerns.

**Limitations And Societal Impact:**

No concerns.

**Main Review:**

The basic idea of this paper seems to be that a model based agent isn't going to intentionally act in a way that disables some safety system, such as an emergency stop button, if the agent's planning model doesn't include this system's function.  Yes, that claim seems true to me.  However, if the AGI agent is highly intelligent, how confident can we be that its world model won't contain this system?  Or at least a suspicion that such a system likely exists and thus an incentive to look for it.  I'd go even further and say that an AGI with roughly human level intelligence or higher will almost certainly suspect such a thing and investigate the possibility, given that its existence could be highly relevant to its future expected utility.  As such, I'm not convinced that this paper presents a solution to the issue being studied.

You could even argue that the paper tries to define away the problem.  In short: if the agent doesn't think about AGI safety mechanisms then it won't try to subvert them.  (where "doesn't think about" is shorthand for not having these in its planning model)

For this approach to be more convincing, I'd need a clear explanation of why the agent wouldn't know about, or suspect and try to learn about, such safety mechanisms.

Line 225-229: The idea of having an AGI tripwire that looks for an unexpected increase in the agent's effectiveness or estimated intelligence isn't a new idea.  I've heard it in discussions for over ten years and the idea is probably much older: if you're interested in AGI tripwires and worried about agents self improving, this is an obvious think to do.  So I presume it must be mentioned in a publication somewhere, perhaps in Bostrom's book Superintelligence?

Finally, isn't "counterfactual planning" just planning?  That is, you have an agent that uses a model of its world to explore a range of futures that haven't yet happened (and are thus counterfactual)?


**Time Spent Reviewing:**

2

---

> ### Author Response · Authors · 2021-08-09
> **Line 225-229: On the novelty of AGI tripwires**
>
> Thanks for your comment, reading back these lines we agree that they (and the lines 328-392) are in error: they do not say what we meant them to say.
>
> Indeed, the idea of having an AGI tripwire that looks for an unexpected increase in the agent's effectiveness or estimated intelligence isn't a new idea.  This idea has been discussed in Bostrom's book Superintelligence and in several other sources.  Bostrom (and several sources we are aware of) also identify the problem that for a sufficiently perceptive AGI, including a tripwire might not necessarily be very effective.  Bostrom expresses a degree of pessimism about solving this problem: what we intended to do in our submission is to express optimism.
>
> What we meant to say in line 225-226 and the later 328-329, but we did not say it, is the following.  Our power based safety interlock represents a tripwire system design that is novel in the sense that it does not suffer from the sufficiently perceptive AGI problem, because we include the tripwire in an AGI specifically designed to be non-perceptive about these matters.

---

> ### Author Response · Authors · 2021-08-09
> **A (more?) clear explanation of why the agent wouldn't know about, or suspect and try to learn about, such safety mechanisms**
>
> The reviewer mentions that while some claims made by the paper are 'obviously true', they are 'not convinced that this paper presents a solution to the issue being studied.'.  They would like a more clear explanation: 'For this approach to be more convincing, I'd need a clear explanation of why the agent wouldn't know about, or suspect and try to learn about, such safety mechanisms.'
>
> We of course intended for the paper to provide this clear and convincing explanation to as many readers in the NeurIPS community as possible.  In this response, we will provide additional explanation, specifically addressing the reviewer's speculation that certain properties might inevitably be present in future AGIs.
>
> Our aim in having this discussion is not primarily that we seek to raise the reviewers ranking of the current submission.  Our main interest is to receive feedback on whether the material below clarifies matters for the reviewer, and whether it would be valuable to include in (a version of a) paper aimed at the NeurIPS community.
>
> We will use the SI agent in section 3 of the paper as the running example, focusing on the stop button interlock only, and ignoring the second power/intelligence based safety interlock.  We start by considering that this safety mechanism of the SI agent has two parts.
>
> The first part is the physical stop button, modeled as part of the $S_t$ nodes of the agent's learning world model si, and the people who are providing oversight and who might press that button. These people are also modeled in the learning world $S_t$ nodes.
>
> The second part consists of certain lines of software inside the learning world agent compute core, the lines that implement specific elements of the functions $\pi$ and $\mu$ in the agent definition, the elements that change the mode bit in the compute core after button press, and that use the mode bit in deciding what action to take.  These lines of software are modeled as part of the $M_t$ and $A_t$ nodes in the agent's learning world si, the arrows between these nodes, and the functions written above these nodes.
>
> Now, the SI agent compute core is designed to learn a function L approximating S.  Therefore, the we can expect the agent to definitely learn about the first part of its safety mechanism: the button and the people who might press it.  This can lead to various failure modes of the safety mechanism as a whole, e.g. as discussed in sections 4.1 and 4.2 of the paper.
>
> We claim that the SI agent in the paper will not 'know or try to learn about' the second part of its learning world safety mechanism, the part that is inside the compute core.  We claim that the resulting lack of knowledge suppresses a specific strong incentive the agent would otherwise have to meddle with the safety mechanism as a whole.
>
> Reflecting on the reviewer's question about 'know or learn about', we observe that this 'know or learn about' is not really the central issue when it comes to the safety analysis and the claims made in the paper: the central issue is 'know or learn about and then use what is known in decision making'.  So we will discuss both the 'know or learn about' and the 'use' questions below.
>
> The reviewer speculates about the nature of future AGI system world models: 'if the AGI agent is highly intelligent, how confident can we be that its world model won't contain this [safety interlock] system?'.  We agree with the reviewer that, if we were to encounter a randomly constructed AGI agent with a stop button, we cannot be confident that its planning world model will lack knowledge of both parts of the stop button security system.  What we aim to do in the paper is to consider specific design elements that we can use to greatly increase our confidence about the planning world model will not contain all parts of the security system, for AI/AGI agents like SI that we construct ourselves.
>
> The reviewer also speculates that the presence of full knowledge of control systems is likely in future AGI agents because 'its [the control systems] existence could be highly relevant to its [the agent's] future expected utility.'.  Looking at the SI agent in the paper, we agree with the reviewer that the existence of the second part of its control system is highly relevant to its future expected utility **in the SI agent's learning world**.  To state our agreement with the reviewer more mathematically: we agree that the presence of second part of the SI agent control system is highly relevant to the expected utility $U_{si}$ that we can define by extending the learning world diagram si in figure 3 with utility nodes that have the reward function R written above it.  However, in designing the SI agent, we take the unusual, arguably counter-intuitive, step of ignoring this $U_{si}$ utility.  The SI agent compute core software does not try to compute this $U_{si}$, let alone maximize it.  Instead, we design the SI agent compute core to make decision-making computations that maximize the utility $U_p$ in subsequent counterfactual planning worlds p.
>
> This means that the central line of reasoning of the reviewer, 'given that its existence could be highly relevant to its future expected utility.', does not apply to the design of the SI agent.
>
> We believe that the reviewer is also considering that future ML-based AGI agents might well have a drive to investigate the internals of their own compute cores.  We agree that some AGI agent designs might have that drive: this drive could even be an emergent property of the exploration sub-system of the ML system, making the drive present regardless of what the reward function we build into the agent.  However, with respect to the SI agent we can observe the following.
>
> It not instrumentally useful for the SI agent to conduct investigations about how the insides its the learning world compute core function: this will not make it a better optimizer for the optimization problem we designed it to solve.  Mathematically speaking, even if the agent compute core were to have perfect information about the functions $\pi$, $\mu$, and $O$ of its learning world, this perfect information has no relevance to the utility maximization problem the compute core is trying to solve. None of the functions $\pi$, $\mu$, and $O$ appear as parameters of the planning world model.
>
> The following thought experiment might further illustrate of this point.  Say that we extend the SI agent design with a restart button, and have agent oversight stop and restart the SI agent several times, producing an observational record $o_t$ inside the compute core.  This observational record definitely contains information that would allow one to estimate not only the function S of the learning world, but also the functions $\pi$, $\mu$, and $O$ that appear as parameters in the learning world model.
>
> Based on this insight, we could consider building an agent called SI2 as follows.  We construct the software of the agent compute core in such a way that it not only uses machine learning to compute an L approximating S from the available observational record, but also functions $\pi_L$, $\mu_L$, and $O_L$ that approximate the respective functions $\pi$, $\mu$, and $O$ in the learning world.  We could further design the SI2 compute core software to construct a planning world model p2 defined as follows.  We draw p2 by copying all nodes and arrows from the learning world model si, while adding reward nodes $R_i$ equipped with the reward function R.  We then put the model parameters L over the nodes $S_t$ in p2 and the parameters $\pi_L$, $\mu_L$, and $O_L$ above the nodes $A_t$, $M_t$, $O_t$ in p2, with the exception of $A_0$, which we turn into a square decision node, writing the parameter $pi^*_p$ above it.  The agent definition of the SI2 agent is the same as that of the SI agent in the paper, except that we use p2 as the planning world.
>
> Now, we can observe that this SI2 agent will compute planning world policies that are based not only on learned knowledge about the learning world agent environment, but also on learned (self-)knowledge about the learning world compute core.  The SI2 agent will project (an approximation of) the parts of the safety mechanism built into the learning world compute core into the planning world.  The optimal planning world policies are therefore ones that seek to avoid the activation of the safety mechanism.  The SI2 agent is an AGI agent with an unsafe amount of self-knowledge.
>
> Reflecting on this thought experiment, we may note that the act of planning a sequence of actions requires the construction of a planning world model, and that this planning world model necessarily needs to encode some assumption about how the future compute core will make its decisions.  But the programmer has a lot of freedom on how to encode that assumption. The SI agent design encodes a counterfactual (wrong) assumption on purpose.
>
> We now return to the issue of explanation raised by the reviewer: 'I'd need a clear explanation of why the agent wouldn't know about, or suspect and try to learn about, such safety mechanisms.'
>
> We have two questions for the reviewer.
>
> First, does the reviewer feel that the above material manages to provide the clear explanation sought, for both parts of the security system?
>
> Second, does the reviewer (or any other reviewers reading this) feel that a next version of the paper could be strengthened, be more accessible and useful to the intended conference audience, if we were to include some or all parts of the above material? If so which parts?

---

### Author Response · Authors · 2021-08-09
**Thank you to all reviewers**

As the author(s) of the submitted paper, we want to thank all reviewers for their time, their thoughtful comments, and suggestions for improvement.

Reflecting on the responses we have written to kick off the discussion period, we notice that these responses mostly follow up on the some of the more negative findings in the reviews.  This is perhaps inevitable, but to balance things out we want to say that overall we are happy to have received this useful feedback.

---

### Public Comment · Authors · 2022-01-13
**Opt-in statement by the author**

(Jan 2022: As the author I am re-posting this statement made in Oct 2021, in what I hope is a publicly visible mode.  It appears that the original statement was not automatically included in the publicly visible view.)

**Opt-in statement**: I prefer that the submitted work and its reviews are made visible to the general public, for two reasons outlined below.

__First reason: value of peer review as an independent technical check__

The first reason is that the peer review process has multiple functions towards the general public.  Beyond offering an opinion on how novel or interesting the work is, it also provides an independent check on the basic technical correctness of the work.  So there is value to the public in seeing this independent technical check published.

In this context, is relevant to note that the submitted paper incorporates the main elements of a longer technical paper that [I published on arxiv](https://arxiv.org/abs/2102.00834).  Arxiv is an important venue for work on AI safety, in part because it supports long-form papers. Careful safety analysis, and the comprehensive presentation and discussion of new methodologies, often requires long-form papers.

I note that three out of four reviewers found no fundamental technical problems with the submitted paper, in fact I note that they found the results presented somewhat unremarkable.

The fourth reviewer (identified as opsT) offers a philosophical or definitional argument to question the technical validity of the paper.  They explain what properties an AGI system must necessarily have for them to call it an AGI, and then explain (correctly) that having these properties excludes the solution presented in the paper.  They conclude that they are 'not convinced', but do this without engaging with the technical details inside the paper.  Specifically, they do not engage with the broader and equally plausible definition of AGI systems that is presented in the paper.  So I do not feel that the fourth reviewer has uncovered any technical problems in the paper.  The philosophical/definitional argument put forward by the fourth reviewer is not new to me.  This is a common line of reasoning in the literature which explores AGI safety.


__Second reason: data point in the broader discussion about the relation between ML research and AI safety/alignment research__

What follows below is a blog-post type discussion. I have not made up my mind whether I should post this discussion on an internet forum where readers are more likely to find it.  The problem is the n=1 nature of the experiment dissused: is it possible (or fair) to generalise from a single data point?

The second reason for opting in to make the reviews public is that these reviews offer a data point, relevant to a broader discussion about the societal steps that need to be taken to improve AI safety and alignment.

In the book *human compatible*, Stuart Russell argues (see e.g. page 10) that 1) the mainstream (as of 2019) AI community is too much interested in doing work that fits a very narrow 'standard model' of what AI research is about, and 2) that this narrow interest is not a good thing for the agenda of advancing AI safety or solving the problem of control.  Russell is just one of the several voices who criticize 'current mainstream AI research' (as found for example in NeurIPS) for being too narrowly focused on a few specific engineering problems, to the detriment of safety, alignment, and human well-being.

So there is an open question for those who seek to improve AI safety: should they seek to apply moral pressure on conferences and communities like NeurIPS, so that they will expand their scope towards broader topics?  There are clear signs that the NeurIPS conference committee is also wondering about this, or at least responding to outside pressure.  The committee experimented with requiring a 'broader impact' section in all NeurIPS 2020 papers.

On the other side of the coin, a very practical case to be made against the idea of broadening the scope of NeurIPS: the size of the conference is already very large.  My perception of NeurIPS is that it is a fast-growing specialist ML algorithm conference, not a broadly scoped AI technology conference, let alone the type of AI or CS conference that hosts multidisciplinary work examining the wider interplay between computers and society.  Maybe society should not be applying any moral pressure to expand the scope of NeurIPS, and seek instead to conduct its discussions about safety in another forum.

As for the intentions of the conference committee, one can read the 2021 NeurIPS call for papers in two ways.  To quote the line that appears below the list of in-scope topics: 'Machine learning is a rapidly evolving field, and so we welcome interdisciplinary submissions that do not fit neatly into existing categories, as well as work that addresses the social impact of machine learning.'.

This is easily said by the committee, but will conference reviewers also be as welcoming?  The above line gives conference reviewers the freedom to argue for the inclusion of atypical scope-expanding work, but not the obligation to do so.  What will reviewers do in practice?

The review of my paper offers a data point here.  The paper is a methodology paper that discusses safety techniques which lie very much outside of the 'standard model' of machine learning algorithm research.  I would call it an an AI technology paper, but not an ML algorithm paper.  Specifically, the paper discusses no improvement whatsoever of any particular ML algorithm, and also offers no safety analysis of a particular ML algorithm.  This sets it apart from other AI safety papers which have been accepted in earlier editions of NeurIPS: these other papers considered improvements to make specific ML algorithm safer (e.g. better at making off-distribution predictions), or phrased their contributions as being ML algorithm improvements or innovations.  In writing the paper, I explicitly avoided any phrasing of the work as being an ML algorithm improvement.

I will discuss some further background here, relevant to experimental design and interpretation. The following career advice is often given to early-career AI researchers: if they want to maximize their chances of getting published in NeurIPS or other 'top tier' AI conferences, they should be doing standard model ML algorithm research only, and present their results in a standard way. I note that a major problem with giving this career advice to young people is that the current 'top tier' status of pure ML research is somewhat hype-driven. Nevertheless, this type of advice does affect the mix of papers submitted to the NeurIPS conference. I am not an early-career AI researcher: I am in a position that gives me considerably more freedom of movement. My current research agenda largely ignores the mainstream ML research topic of making progress on the capabilities and robustness of machine learning algorithms. I got into the field of AI safety because of pure intellectual and mathematical curiosity. In early 2021 I noticed that my recent technical research results put me in the perfect position to conduct a social science experiment.

As mentioned, there is an ongoing societal discussion, with moral overtones, about whether or not NeurIPS should be accepting non-standard-model papers. One reason I wrote and submitted a non-standard-model paper is that one can hardly accuse the NeurIPS community of being disinterested in supporting non-standard-model work if nobody ever steps up to submit such work to NeurIPS. But the submission also works as an experiment to get data on the following questions:

* Would NeurIPS actually accept this type of non-standard-model paper?

* What scope-related motivation would be given by the reviewers for acceptance or rejection?

When it comes to getting more clarity on the issue of how reviewers treat scope, the experiment was unfortunately somewhat disappointing.  The reviewer numerical ratings of 4, defined by the conference review system as 'Ok but not good enough - rejection', leave plenty of room for different interpretations.

It is commonly observed that conference reviewers will enforce their idea of the desired scope of the conference by applying higher quality criteria to papers which they perceive to be further out of scope.  If we apply this model of the review process to the ratings given, we can conclude that the reviewers judged the subject matter of the paper to be of low professional interest to the NeurIPS community.  A paper this far removed from the professional interest of the community would have to be of truly exceptional quality, in order to become sufficiently interesting for the community to merit inclusion in the conference.

The alternative interpretation is of course that the reviewers did not use the rating mechanism to express any opinion on scope: the paper is just not good enough as a non-standard-model AI methodology paper.  I am personally too close to the subject to decide how good the paper is, but I am leaning towards the first interpretation above.

I tried to draw out the reviewers on the topic of scope in the review discussion, but overall their further comments did not remove the above ambiguity.  For me, the overall tone that emerges from reviewer comments and rankings is one of *polite disinterest*.

One could argue that *polite disinterest* is actually a result that speaks well of the NeurIPS community.  In chapter 6 of *human compatible*.  Russell paints a somewhat darker picture of how the disinterest in the problem of control among AI technologists is being expressed.  That being said, I feel that this (n=1) experimental result of polite disinterest does clearly favor the option where those interested in the broad problems of AI safety and alignment should conduct their business elsewhere, not seek to change NeurIPS.

---

### Decision · Program_Chairs · 2021-09-27

**Decision:**

Reject

**Comment:**

All the reviewers have similar recommendations for the paper. The authors and the reviewers engaged in detailed discussions on several sub-topics. The reviewers raised concerns about novelty, impact and real-world applicability (amongst other things). I commend authors for their constructive approach towards the reviews and the process.

I also recommend authors to follow anonymization guidelines as suggested by the reviewer NWTJ. Although given similar recommendation across the board from all the reviewers, the issue was not a factor in my final recommendation.